Activity strategy and pattern of the Siberian jerboa (Orientallactaga sibirica) in the Alxa desert region, China

Ji Yu 1 2 3
Yuan Shuai 1 2 3
Fu Heping fuheping@126.com 1 2 3 4
Yang Suwen 1 2 3
Bu Fan 1 2 3
Li Xin 1 2 3
Wu Xiaodong wuxiaodong_hgb@163.com 1 2 3
1 College of Grassland, Resources and Environment, Inner Mongolia Agricultural University , Hohhot , China
2 Rodent Research Center, Inner Mongolia Agricultural University , Hohhot , China
3 Ministry of Education Key Laboratory of Grassland Resources , Hohhot , China
4 College of Life Sciences, Inner Mongolia Agricultural University , Hohhot , China
Harpur Brock
Electronic publication date: 2021 Mar 10
Publication date: 2021
Volume: 9
Electronic Location ID: e10996
Received 2020 Apr 20; Accepted 2021 Feb 2
Copyright: ©2021 Ji et al.
Copyright year: 2021
Copyright holder: Ji et al.
License: This is an open access article distributed under the terms of the Creative Commons Attribution License, which permits unrestricted use, distribution, reproduction and adaptation in any medium and for any purpose provided that it is properly attributed. For attribution, the original author(s), title, publication source (PeerJ) and either DOI or URL of the article must be cited.
License URL: https://creativecommons.org/licenses/by/4.0/

Keywords: Activity pattern, Activity strategy, Infrared camera, Jerboa

Funding: The National Natural Science Foundation of China 32060256 32060395 31772667 The Natural Science Foundation of Inner Mongolia 2018MS03014 2019MS03012 The Key Project of the Education Department of Inner Mongolia NJZZ17055 The Ministry of Education Key Laboratory of Grassland Resources This work was supported by the National Natural Science Foundation of China [grant numbers 32060256, 32060395, 31772667], the Natural Science Foundation of Inner Mongolia [grant numbers 2018MS03014, 2019MS03012], the Key Project of the Education Department of Inner Mongolia [grant number NJZZ17055], and the Ministry of Education Key Laboratory of Grassland Resources. The funders had no role in study design, data collection and analysis, decision to publish, or preparation of the manuscript.

==============================
Rodents exhibit seasonal changes in their activity patterns as an essential survival strategy. We studied the activity patterns and strategies of the Siberian jerboa (Orientallactaga sibirica) in the Alxa desert region to better understand the habitats and behavioural ecology of xeric rodents. We conducted an experiment using three plots to monitor the duration, time, and frequency of the active period of the Siberian jerboa using infrared cameras in the Alxa field workstation, Inner Mongolia, China in 2017. The relationships between the activity time and frequency, biological factors (perceived predation risk, food resources, and species composition), and abiotic factors (temperature, air moisture, wind speed) were analysed using Redundancy Analysis (RDA). Our results showed that: (1) relative humidity mainly affected activities in the springtime; temperature, relative humidity and interspecific competition mainly affected activities in the summertime; relative humidity and perceived predation risk mainly influenced activities in the autumn. (2) The activity pattern of the Siberian jerboa altered depending on the season. The activity of the Siberian jerboa was found to be bimodal in spring and summer, and was trimodal in autumn. The activity time and frequency in autumn were significantly lower than the spring. (3) Animals possess the ability to integrate disparate sources of information about danger to optimize energy gain. The jerboa adapted different responses to predation risks and competition in different seasons according to the demand for food resources.

Introduction

The activity patterns of animals indicate various evolutionary adaptations. Each population has a seasonal activity pattern that is best suited to local conditions. Individuals optimizing their activity patterns have the most significant advantage in natural selection (Kronfeld-Schor & Dayan, 2008; Hemami et al., 2011; Mastureh et al., 2017; Mansureh & Morteza, 2018). The activity pattern of animals is a comprehensive adaptation to the periodic changes of various environmental conditions, including non-biological conditions such as light, temperature, humidity, food need, intra-species community relations, and natural enemies (Halle, 2000). Activity patterns are also influenced by predation risk (Orr, 1992; Fenn & Macdonald, 1995), competition (Alanara, Burns & Metcalfe, 2001; Kronfeld-Schor & Dayan, 2008), food availability (Orpwood, Griffiths & Armstrong, 2006), reproductive status (Schrader, Walaszczyk & Smale, 2009), nutritional status (Metcalfe & Steele, 2001), habitat (Wasserberg, Kotler & Abramsky, 2006), and physical factors (Fraser, Metcalfe & Thorpe, 1993). Temperature affects animal activity patterns, and the effects of temperature on behaviour and its interactions with other factors have been experimentally studied (Levy, Dayan & Kronfeld-Schor, 2007). Research has indicated that the activity period of animals may change with seasonal temperature variations (Lee et al., 2010). Activity periods may also vary between microhabitats with different wind speeds (Melcher, Armitage & Porter, 1990). Air moisture is particularly important for animals living in warm or hot environments due to the influence of air moisture on heat balance (Kausrud et al., 2018; Shuai et al., 2014). The availability of food is the primary factor influencing changes in the activity patterns of rodents. Studies have shown that ecological factors directly related to energy demand affected activity patterns of animals (Denis, Susan & Donald, 1980). Predation risk may be another ecological factor affecting activity patterns (Claire & Ferrando, 2017). Although there are many findings related to factors that influence animal activity patterns, most studies have focused on only one aspect and have failed to consider the relative importance of other factors (Shuai et al., 2014; Kei & Motokazu, 2017). The species composition, including the presence or absence of potential competitors (Elke et al., 2013), of the rodent community also influences the nocturnal activity patterns of rodents. We sought to determine the main factors affecting the activity of the Siberian jerboa.

Studies have found that rodent activity patterns in desert areas change with the seasons (Gregory et al., 2001; Richman & Van De Graff, 1973). For example, golden spiny mice (Acomys russatus) shown diurnal activity patterns with a midday peak in winter and a bimodal pattern with peaks in morning and afternoon in the summer (Shkolnik, 1971; Kronfeld-Schor & Dayan, 2008). It has been speculated that the less typical diurnal pattern demonstrated by A. russatus is due to competitive displacement (Kronfeld, Shargal & Dayan, 1996). The kangaroo rat (Dipodomys merriami) responds to winter by decreasing its activity in November (Richman & Van De Graff, 1973). Seasonal shifts in activity patterns have also been observed in studies of other rodent species, and analysis of daily activity rhythms have shown that the number of peak periods of activity are higher during high-temperature seasons than that in low-temperature seasons. Research of monthly activity rhythms demonstrated that activity was higher in high-temperature months than in low-temperature months. For example, the activity pattern of Verreaux’s sifaka (Propithecus verreauxi) was bimodal during the low-temperature winter period to reduce energy loss, and unimodal with more extended activity periods in high-temperature summer (Erkert & Kappeler, 2004). The Japanese flying squirrel (Pteromys momonga) is bimodal in temperate seasons and trimodal in cold seasons. This species reduces its active time in the cold season to reduce energy consumption caused by long-term exposure to low temperatures (Lee et al., 2010). Rodents alter their activity patterns when external environmental factors change with the season. Variations in the external biological and abiotic environment cause the transformation of animal activity patterns for reasons of survival, and animals usually adjust their activity frequencies and times to cope with these changes (Delany, 1972).

A pattern has emerged in some studies to show that abiotic environmental factors such as temperature often provide a range within which rodents are active. The quantity or quality of food will determine activity levels within this range (Bartholomew & Cade, 1957). Emlen (1996) and MacArthur & Pianka (1966) created the optimal foraging theory, which attempts to explain and predict many aspects of animal foraging behaviour. This theory supposes that the foraging behaviours and adaptability of animals are maximized through natural selection, but are subject to certain restrictions. The optimal foraging theory predicts that risk-taking decisions vary with the perceived threat level (Kelly, Cypher & Germano, 2020) and that animals have to face trade-offs when they encounter predators when foraging. Biological factors such as food resources and predation risk may change the survival strategy that an animal will adopt. An animal’s response to predatory pressure may be to remove itself from the predator’s foraging microhabitat (predator avoidance mechanism), or to reduce the probability of successful predation within the predator’s perceptual domain (anti-predator mechanism). Predator avoidance mechanisms are typical patterns of behaviour exhibited by animals; these mechanisms include occupying (e.g., cover or dense vegetation), changing their foraging habitats (spatial avoidance), or adjusting their activity periods (temporal avoidance). A variety of morphological and behaviour traits represent anti-predator mechanism (Brodie, Formanowicz & Brodie, 1991). The ability of animals to bear the risk of predation is related to their own characteristics. An animal with a higher basal metabolism is more sensitive to the risk of predation. When animals forage under conditions of food shortage, they will change strategy from risk-aversion to risk-proneness (Wei et al., 2004a; Wei et al., 2004b).

The Siberian jerboa (Orientallactaga sibirica) (Michaux & Shenbrot, 2017) is a rodent species found predominantly in the desert and semi-desert of Alxa. It enters into hibernation in early September and emerges from hibernation in late March or early April of the following year (Li & Han, 1990; Zhou et al., 1992). The Siberian jerboa is usually active in the evening and before dawn, and is not easily found in daylight hours (Liang & Xiao, 1982; Dong, Hou & Yang, 2006). We sought to determine the activity pattern of the Siberian jerboa, its impact factors, whether its activity pattern changes with the seasons and the reason for the shift. We also investigated the survival strategies involved in the variations of activity.

We selected biological factors (perceived predation risk, food resources, and species composition of rodent community), and abiotic factors (temperature, air moisture, wind speed) as possible influencing elements for the jerboa activity patterns to explore whether the Siberian jerboa had the same seasonal variations as other rodent, and to determine the survival strategies and other factors behind these variations. We proposed the following hypotheses: (1) In spring, after a long hibernation period, energy supply is incredibly important and food resources may be an essential factor influencing the activity pattern of Siberian jerboa. In summer, high temperatures may limit the activity of the jerboa, but to compensate for this thermal constraint, wind speed, and air humidity are influential. Food resources become critical in autumn to store the energy needed during the long hibernation period. (2) There were seasonal changes in the activity patterns of the Siberian jerboa. The dominant factor driving this shift may be competition or temperature. (3) The mechanism driving this shift is risk-taking decisions should vary in response to perceived levels of threat.

Materials and Methods

Study area

Our study was conducted in the southern part of Alxa Zuo Qi at the eastern edge of the Tengger Desert, Inner Mongolia, China (E104°10′–105°30′, N37°24′–38°25′) from April to October, 2017. The area has a continental climate with cold, dry winters and warm summers. Annual temperatures range from -36 to 42 °C with a mean of 8.3 °C. Annual precipitation ranges from 45 to 215 mm with approximately 70 percent of precipitation falling from June to September. The potential evaporation range is from 3,000 to 4,700 mm, and the annual frost-free period is 156 days. The Siberian jerboa is a common species distributed throughout the study site. Approximately 5–15% of the ground is covered with shrubs, forbs, and some gramineous plants. Shrubs mainly consisted of Zygophyllum xanthoxylon, Nitraria tangutorum, Caragana brachypoda, Ceratoides latens, Oxytropis aciphylla, Artemisia sphaerocephala, and Artemisia xerophytica with Reaumuria soongorica as the dominant species. The major grasses/forbs species were Cleistogenes squarosa, Peqanum nigellastrum, Cynanchum komarovii, Salsola pestifer, Suaeda glauca, Bassia dasyphyll a, Corispermum mongolicum, Artemisia dubia, and Plantago lessingii (Yuan et al., 2018). Coexistent rodent species included Dipodidae (Orientallactaga sibirica, and Dipus sagitta), Cricetidae (Meriones meridianus, Cricetulus barabensis, and Phodopus roborovskii) and Sciuridae (Spermophilus alaschanicus). The natural enemies of rodents in the area are corsac (Vulpes corsac), eagle owls (Bubo bubo), and snake (Agkistrodon halys). According to our field observations, snakes, eagle owls, and foxes are active at night. The spring season is from March to May, summer is from June to August, autumn is from September to November, and winter is from December to February of the following year according to the climate characteristics of the test site.

Camera trapping

We deployed camera traps in three plots, each approximately 1 hm2 in area, each separated by more than 500 m. We deployed a survey infrared camera (Infrared monitor E1B, Lianyungang Jinsheng Technology Co., Ltd., China) to each plot. All cameras were active in May (spring), July (summer), and September (autumn) each year. Since the Siberian jerboa is a nocturnal rodent, the cameras were active from 19:00 to 07:00 the next day, divided into 12 time periods (i.e., 19:00–20:00, 20:00–21:00, and so on). Cameras functioned for four consecutive days each month for a total of 36 trap nights. Each camera was placed on a stake approximately 30 cm above the ground facing a lure 1.5–2.0 m away. Vegetation in the camera’s line of sight was cleared to prevent false triggers. A peanut (Arachis hypogaea) was used as the lure. A pre-experiment check was conducted on the placement of the infrared cameras. Infrared cameras were randomly placed in the territory of the Siberian jerboa to find out where this rodent was active and we selected the active sites for our observations. The camera parameters were set to shooting mode (video), and video was recorded for 2 min when triggered, with no quiet period between trigger events. We checked the performance of the camera and replaced the battery and storage card every morning when data was collected. The videos were downloaded to a computer and each camera was assigned a point number. We identified each wildlife video and entered information on each video into an Excel table according to the camera number and appearance time to avoid repeatedly counting the activity information of the same animal in a short time. Multiple videos of the same animal within 30 min are entered as one record. We identified each photo of an animal for its species, recorded the time and date, and rated each photo as a dependent or independent event. An independent event was defined as the number of distinctly different individuals of a species detected within a 30-min period (Di Bitetti, Paviolo & Angelo, 2006; Davis, Kelly & Stauffer, 2011; Murphy et al., 2016). As we were unable to discriminate individual small mammals, each detection event was noted as one animal, unless there were multiple animals in the images. We defined an independent event as (1) consecutive photographs of different individuals of the same or different species, (2) consecutive photographs of individuals of the same species taken more than 0.5 h apart, (3) non-consecutive photos of individuals of the same species (O’Brien, Kinnaird & Wibisono, 2003; Duquette et al., 2017). We watched the video and recorded the duration of each appearance of jerboa and summed up the period of each appearance of an animal within a 60 min period, which was recorded as the activity time (O’Brien, Kinnaird & Wibisono, 2003). Vigilance behaviour was defined as a series of physical action response behaviours exhibited by animals in response to existing or potential risks in the surrounding environment (Wang, Hai & Wang, 2015). It can be divided into the following cases: (1) Tweet: when danger is detected, a sitting, standing, or squatting posture is used to look directly at the threat and a series of screams are emitted to warn other members of the same species; (2) Alert: interruption of the ongoing behaviour (such as running, feeding, foraging, etc.), assume the squat posture, static, or accompanied by a rapid head twist to observe the surrounding environment to determine whether there is danger around, generally for no more than 3 s; (3) Watching: observe the movement of the surrounding environment by standing, sitting, or squatting, and the field accompanied by the head writhing for a longer time than 3 s; (4) Avoiding: interruption of the ongoing behaviour (such as running, feeding, foraging, etc.) when danger is detected, or a call is heard, to quickly run back to the den, sometimes accompanied by a cry of alarm.

Abiotic factors

Climate data were collected from the Luanjingtan Weather station of Alax, and the average distance from the study area was 5.5 km. The climate data reflected the local environmental conditions in the study plots (Wu et al., 2016). We collected hourly data for temperature, relative humidity, and wind speed from the China Meteorological Administration (http://www.cma.gov.cn/) over the same four days that the camera operated each month.

Biological factors

Food resource

We conducted vegetation sampling in the three plots in May, July, and September in 2017. We randomly placed three 100-m2 square sampling plots on each treatment unit to sample shrubs and randomly placed three 1-m2 quadrats in each 100-m2 square plot to sample grasses and forbs. Three shrubs of each species were randomly selected from a shrub sample of 100-m2, their crowns were measured and an appropriate amount of the aboveground part was taken. Samples were dried and the dry weight was taken. We measured the height of herb samples taken from a 1-m2 plot, cut the herbs and dried them, then took the dry weight (Yuan et al., 2018). We estimated the aboveground standing biomass of shrubs, grasses, and forbs by species. Siberian jerboa are known to feed on the green parts of plants, the underground parts of plants, and seeds and insects in approximately equal proportion (Shenbrot et al., 2008). The same species distributed in different regions may also have different diets. We conducted feeding behaviour experiments of the Siberian jerboa were conducted in 2017 to determine the food resources in the study sites. The experiment was conducted as follows: Jerboas were live-trapped for each season (spring, summer, and autumn) from the desert habitat at a specified distance from the study sites and were fasted for 8 h before placing them into cages at dusk. A total of 14 jerboas were used (two males, and two females in May, three males and two females in July, three male and two female jerboas in September), weighing on average 95.80 g ± 17.74 g (mean ± SD). Each jerboa was randomly assigned to one cage. We provided each plant species to each subject in 100 mm petri dishes placed in a randomized array with one species per dish. We fed jerboas every 3 h, 2- 3 times per night. Ten to twelve plant species were fed in each feeding. The plant species fed to each jerboa was the same on the same night, but plant species were arranged randomly for each jerboa. We collected the remaining plants (including those cached throughout the cages) and plant remnants, and separated and weighed them by species when putting the new plant species in. The same plant species with same weight were placed outside of the cages as a control group to determine water loss. We calculated species composition of consumed plants by subtraction: Y=A−B1−E, where Y is mean food consumption, A is mean initial weight, B is mean remaining weight, E is mean rate of water loss.

Preference index (PI) was calculated according to the daily food consumption of each plant by the formula: PI=RIRB, where PI is mean preference index, RI is mean mass percentage of a plant’s consumption in total food consumption, and RB is mean mass percentage of a plant in total feed. The plant species was chosen by calculating the total food biomass and preference food biomass. Preferred foods were selected using the preference index (Batzli & Pitelka, 1983). Preferred food biomass represented the food resources in the habitat.

Perceived predation risk

Vigilance time, vigilance frequency, and the distance of the vigilance alert are three indicators for assessing the perceived predation risk level of small rodents (Wan, 2019). Vigilance behaviour is one of the most essential countermeasures against predation, which depends heavily on the perceived predation risk (Limaa, 1987). Studies have shown that when the risk of predation increases, an animal’s time-allocation strategy changes, reducing the risk of predation by increasing alertness, reducing foraging and other behaviors, and vice versa. It is believed that there is a trade-off between the alertness of animals and the activity intensity of other behaviors, such as foraging. This theory is called the predation risk allocation hypothesis (Wei et al., 2004a; Wei et al., 2004b; Steven & Peter, 1999) and it is believed that the higher the risk in the habitat, the greater the proportion of vigilance in total activity. To assess perceived predation risk, we measured the proportion of vigilance frequency in total activity frequency. We evaluated the perceived predation risk of Siberian jerboa by vigilance behavior, and verified our results using the vegetation structure and by excluding the influence of other factors on vigilance behavior.

Studies have shown that safety for small mammals is correlated with some measure of vegetation density, such as shrub coverage or grass height (Jacob & Brown, 2000; Morris & Davidson, 2000). Changes in vegetation may change an animal’s perceived risk by increasing a potentially risky structure (Hagenah, Prins & Olff, 2009). Small-scale changes to the vegetation structure have been shown to alter the fear levels of prey, regardless of the abundance of predators (Wheeler & Hik, 2014), and may influence the perceived predation risk more than actual predator cues. Reductions in ground cover, grass height, and horizontal structure may increase the perception of risks (Banasiak & Shrader, 2015); Shraderetal2008. We used grass height and shrub coverage to assess predation risk. The calculation formulas are as follows: (1) AH=LH+MH+SH3

AH represents the average height (cm) of a shrub species. LH, MH, and SH represent the height (cm) of large, medium, and small plants of the shrub species. (2) C=3.14×SR2×Den100m2×100%

C represents the average coverage per unit area of a shrub species (%), and SR represents 1/2 (m) of the average canopy width of the shrub species. (3) TC= ∑I=1SCi

TC represents the total coverage per unit area of the shrub (%), S represents the number of species, and Ci represents the coverage per unit area of the ith shrub (%).

The vigilant behavior of animals is related to the risk of predation and the trade-off between vigilant and predation. Vigilance is affected by other factors, including an animal’s sex and age (Randall, Konstantin & Debra, 2000; Xia et al., 2011), population size (Xia et al., 2011; Tchabovsky & Sergei, 2001), individual position in the group, and environmental characteristics (Bekoff, 1995). Therefore, the vigilant behavior of animals is the result of the comprehensive effect under the influence of multiple factors (Bekoff, 1995). We excluded any coexisting nocturnal rodents when considering the vigilance behavior of our test species.

Spearman correlation analysis was used to analyze perceived predation risk, shrub coverage, and grass height (Table 1). Perceived predation risk was significantly negatively correlated with shrub coverage and grass height in spring and summer (P < 0.01). Perceived predation risk was significantly negatively correlated with grass height (P < 0.01), and negatively correlated with shrub coverage. Some studies of the desert jerboa have shown that certain species of the dipodidea prefer bare land with low vegetation coverage (Brown, 1980; Mansureh & Morteza, 2018). We found that vigilance behavior may represent the perceived predation risk of Siberian jerboa to some extent, depending on the vegetation structure. Therefore, we measured the perceived predation risk of Siberian jerboa by the ratio of vigilance to all behaviors.

Table 1 Correlation analysis of perceived predation risk and vegetation structure in different seasons.

Season	Item	Perceived predation risk	Grass height	Shrub coverage	
Spring	Perceived predation risk	1.000			
Grass height	−0.118**	1.000		
Shrub coverage	−0.059**	−0.224**	1.000	
Summer	Perceived predation risk	1.000			
Grass height	−0.282**	1.000		
Shrub coverage	-0163**	−0.018	1.000	
Autumn	Perceived predation risk	1.000			
Grass height	−0.342**	1.000		
Shrub coverage	−0.030	0.055	1.000	
Notes.

* Significant correlation at 0.05 level.

** Significant correlation at 0.01 level.

Spearman correlation analysis was used to analyze vigilance frequency and the relative population number for rodents in different seasons. The results showed no significant correlation between the vigilance frequency and the other three species of rodents co-existing in the same area. The influence of the population size of the Siberian jerboa on its vigilance behavior was also excluded (Table 2).

Table 2 Correlation analysis of Vigilance frequency and the population relative number of rodents in different seasons.

Season	Item	Vigilance frequency	Dipus sagitta	Phodopus roborovskii	Meriones meridianus	Allactaga sibirica	
Spring	Vigilance frequency	1.000					
Dipus sagitta	0.021	1.000				
Phodopus roborovskii	0.014	0.500**	1.000			
Meriones meridianus	0.007	0.500**	−0.500**	1.000		
Allactaga sibirica	−0.021	−1.000**	−0.500**	−0.500**	1.000	
Summer	Vigilance frequency	1.000					
Dipus sagitta	0.040	1.000				
Phodopus roborovskii	–	–	1.000			
Meriones meridianus	−0.040	−1.000**	–	1.000		
Allactaga sibirica	0.115	−0.500**	–	–	1.000	
Autumn	Vigilance frequency	1.0001.000					
Dipus sagitta	−0.060	1.000				
Phodopus roborovskii	–	–	1.000			
Meriones meridianus	0.060	−1.000**	–	1.000		
Allactaga sibirica	−0.100	0.000	–	0.000	1.000	
Notes.

* Significant correlation at 0.05 level.

** Significant correlation at 0.01 level.

We found that it may be possible to evaluate the perceived predation risk of Siberian jerboa by its vigilance behavior.

Species composition of the rodent community

Rodents were live trapped for 4 consecutive days at 4-week intervals from April to October in 2017. Trapping did not occur from November to March. Traps were baited with fresh peanuts and checked in the morning and afternoon each day. The life span of the jerboa is longer than 2 years, and the average life span of non-jerboa species is shorter than 2 years. Each captured jerboa individual was sexed, and was marked with a 1.5 g aluminum leg ring (0.4 cm diameter) with a unique identification number (ID) attached to the left hind foot. Each captured non-jerboa individual was sexed, marked with an electronic chip with a passive integrated transponder (Remex-X003, 2 ×1.8 mm, Guangzhou Ruimai Intelligent Technology Co. Ltd., China) with a unique identification number (ID) injected under the pelage. The passive integrated transponder had a life span of 2 years. The capture station, sex, body weight, and reproductive condition of each individuals was recorded. Males were considered in reproductive condition if they had scrotal testes. Females were considered reproductive if they possessed enlarged nipples surrounded with white mammary tissue, or a bulging abdomen. In order to avoid accidental death, traps were closed on extremely warm or rainy days, and the trapping time was extended after extremely warm or rainy days to ensure 4 days of trapping in each month (Wu et al., 2016). To assess the effectiveness of the aluminum leg rings, we conducted a pre-experiment checks in 2018 and 2019. In April and May 2018, the leg rings and electronic chips were used to mark the jerboa simultaneously, and the loss of the leg rings and the electronic chip was recorded in September of the same year. At the beginning of this pre-experiment, we captured 21 Dipus sagitta individuals and 15 O. sibirica individuals in 2018. Six D. sagitta individuals and seven O. sibirica individuals were recaptured in September of 2019, with no loss of leg rings or chips.

We calculated the population relative number of rodents with a hundred cage capture rate (Wu et al., 2016). The calculation formula used is as follows: (4) P=NH×n×100%

P is the capture rate; N is the number of captured individuals; H is total number of cages; N is the number of consecutive days.

The nocturnal species of rodent coexisting with the Siberian jerboa were the D. sagitta, Phodopus roborovskii, and Meriones meridianus. The relative numbers of the three species were summed up as the number of coexisting species.

Statistical analyses

The activity time (the duration of the active period), activity frequency, predation risk, and food resources in different seasons were analysed by one-way ANOVA using SAS 9.0 software. All data were tested using the Shapiro–Wilk method and were found to be normally distributed (Table 3). Spearman correlation analysis was used to analysed the perceived predation risk, shrub coverage, and herb height. SPSS 21.0 was used for the analysis.

Table 3 Normalization test results.

Season	Item	Sig.	Normalization or not	
Spring	Preference food biomass	0.408	Yes	
Total food biomass	0.142	Yes	
Activity time	0.058	Yes	
Activity frequency	0.072	Yes	
Perceived predation risk	0.063	Yes	
Summer	Preference food biomass	0.674	Yes	
Total food biomass	0.434	Yes	
Activity time	0.158	Yes	
Activity frequency	0.991	Yes	
Perceived predation risk	0.390	Yes	
Autumn	Preference food biomass	0.342	Yes	
Total food biomass	0.161	Yes	
Activity time	0.270	Yes	
Activity frequency	0.200	Yes	
Perceived predation risk	0.056	Yes	

Multivariate analysis was performed with CANOCO 5.0 to explore the relationship between the environmental factors (biological and abiotic factors) and activity time and frequency over the different seasons using Redundancy Analysis (RDA). Detrended correspondence analysis (DCA) with detrending by segments was conducted to analyse the data on activity time and frequency in 2017 and to evaluate the gradient length of the first axis when deciding whether to use linear or unimodal based numerical methods. A Monte Carlo permutation test based on 499 random permutations was conducted to test the significance of the eigenvalues of the first canonical axis.

Results

Food resources

According to the preference index in different seasons, food resources varied among seasons. There were 12 species of plants favoured by Siberian jerboa in spring, 18 species favoured in summer, and 20 species favoured in autumn. There were differences in the feeding habits of the jerboa across the different seasons (Table 4) and significant differences in food resources available in different seasons. The preferred food biomass in autumn was significantly higher than that in spring and summer (F2,24 = 15.67, P < 0.0001). The total food resources in spring were significantly less than that in summer and autumn (F2,24 = 18.16, P < 0.0001). The total food resources were significantly higher than the preferred food biomass in different season (spring F1,16 = 5.04, P = 0.040; summer F1,16 = 38.50, P < 0.0001; autumn F1,16 = 8.91, P = 0.0093) (Fig. 1).

Activity time and activity frequency

The activity time of the jerboa was longer in spring and summer than in autumn (F3,33 = 5.64, P = 0.0078). There were two significant activity peaks in the daily activities, which appeared at 21:00-00:00 and 02:00-04:00, respectively, and were significantly different from other non-peaks times (spring F11,132 = 4.81, P < 0.0001; summer F11,132 = 2.86, P = 0.0022). The difference between the two activity peaks was not significant (spring F4,55 = 0.83, P = 0.5100; summer F4,55 = 0.87, P = 0.4876). There were three peaks of activity time in the autumn, which appeared at 20:00-21:00, 23:00-00:00 and 04:00-05:00, respectively, and were significantly different from non-peaks times (F11,132 = 2.23, P = 0.0165) (Fig. 2A).

There was a significant difference between the activity frequency of the jerboa among three seasons (F2,33 = 10.67, P = 0.0003). There were two peaks of activity frequency in spring and summer. The peaks appear in spring at 21:00–00:00 and 01:00–04:00, and were significantly different from non-peaks times (F11,132=3.71 P < 0.0001). The peaks appeared in summer at 21:00–23:00 and 02:00–05:00, and were significantly different from non-peaks times (F11,132 = 3.84, P < 0.0001). There were three peaks of activity time in the autumn, which appeared at 20:00–21:00, 23:00–00:00, and 04:00–05:00, respectively, and were significantly different from non-peaks times (F11,132 = 3.87, P < 0.0001). (Fig. 2B).

There was a significant difference between the total activity time each season. The total activity time in autumn was significantly shorter than in spring and summer. (F2,33 = 5.64, P = 0.0078) (Fig. 2C). There was a significant difference between the total activity frequency in each season, following the order: spring > summer > autumn. (F2,33 = 1.67, P = 0.0003) (Fig. 2D).

Perceived predation risk

Vigilance behaviour in spring typically occurred between 05:00-06:00. In summer, vigilance behaviour typically occurred between 20:00-21:00. In autumn, vigilance behaviour occurred most often between 04:00-05:00 (Fig. 3A, Figs. 3B, 3C). There were significant differences in the proportion of vigilance behaviour in the total activity periods in spring and autumn (Spring F11,110 = 5.70, P < 0.0001; Autumn F11,103 = 2.93, P = 0.0021), but no significant differences during summer (F11,85 = 1.64, P = 0.1008).

Table 4 Preference index of Allactaga sibirica diet in spring, summer and autumn of 2017 at Alxa Zuo Qi, Inner Mongolia, China.

Bold represents preferred food (index > 1).

Plant species	Spring	Summer	Autumn	
Achnatherum splendens	3.433	0.830	0.342	
Agriophyllum pungens	—	0.556	1.656	
Agropyron mongolicum	0.596	—	—	
Allium mongolicum	0.975	0.283	0.168	
Ammopiptanthus mongolicus	—	0.118	0.608	
Artemisia ordosica	0.401	0.047	0.086	
Artemisia sphaerocephala	—	0.050	0.237	
Artemisia xerophytica	0.134	0.147	0.035	
Asparagus cochinchinensis	—	0.179	1.065	
Astragalus galactites	1.275	3.403	1.306	
Atraphaxis frutescens	—	1.019	0.197	
Bassia dasyphylla	1.004	0.769	0.837	
Caragana brachypoda	2.995	1.864	0.396	
Caragana korshinskii	0.790	0.882	0.912	
Carex Stenophylloides	—	1.079	0.786	
Caryopteris mongholica	—	—	1.247	
Ceratoides intramongolica	0.140	0.242	0.276	
Cleistogenes songorica	—	1.488	3.695	
Convolvulus ammannii	5.172	1.338	1.775	
Corispermum mongolicum	—	—	0.248	
Cynanchum chinense	—	0.785	0.733	
Cynanchum hancockianum	0.326	0.001	0.232	
Cynanchum thesioides	—	1.062	1.481	
Echinops gmelini	—	1.873	2.032	
Eragrostis pilosa	—	3.827	—	
Euphorbia humifusa	—	0.911	1.953	
Halogeton arachnoideus	—	0.383	—	
Haloxylon ammodendron	0.199	0.058	0.660	
Hedysarum scoparium	2.060	0.473	0.231	
Ixeris denticulata	—	3.120	—	
Lepidium apetalum	0.694	—	—	
Lycium ruthenicum	0.395	—	—	
Micropeplis arachnoidea	—	1.561	1.772	
Nitraria tangutorum	0.014	0.000	1.259	
Oxytropis aciphylla	3.018	0.943	0.942	
Panzeria lanata var. alaschanica	1.142	0.463	0.000	
Peganum harmala	0.000	0.617	1.485	
Pennisetum centrasiaticum	3.854	2.280	1.914	
Phragmites australis	2.661	1.662	0.940	
Plantago lessingii	—	—	3.305	
Psammochloa villosa	—	0.292	0.674	
Reaumuria songarica	1.448	1.078	0.245	
Salsola collina	6.661	—	2.429	
Salsola collina	—	2.130	—	
Sarcozygium xanthoxylon	0.757	0.147	1.414	
Scorzonera divaricata	—	—	1.172	
Setaria viridis	—	2.941	1.582	
Sonchus arvensis	—	0.924	1.528	
Stipa glareosa	—	2.012	1.544	
Tribulus terrester	—	4.103	—	

There was a significant difference in daily vigilance behaviour frequency in different seasons (F2,33 = 4.05, P = 0.0268). Perceived predation risk in spring was significantly higher than in autumn (Fig. 3D).

Species composition of the rodent community

Among the rodents co-existing with the Siberian jerboa, the nocturnal species is the D. sagitta, P. roborovskii, and M. meridianus. There was no difference in the catch proportion of D. sagitta and M. meridianus between the seasons (D. sagitta F 2,6 = 1.11, P = 0.3902; M. meridianus F2,6 = 0.41, P = 0.6812). There was a significant difference in the catch proportion of Siberian jerboa between the seasons (F2,6 = 4.85, P = 0.0558). There was a significant difference in the total number of coexisting species between seasons (F2,6 = 7.25, P = 0.0251) (Table 5).

Figure 1 Preference food biomass (±se) and total food biomass (±se) in different seasons.

Different lower case letters indicate significant difference among seasons at 0.05 level. Different capital letters indicate significant difference among preference food biomass and total food biomass at 0.05 level.

Figure 2 (A–B) Activity time (±se) and (C–D) activity frequency (±se) of Siberian jerboa in different seasons.

Different lower case letters indicate significant difference among seasons at 0.05 level.

Figure 3 (A–D) Perceived predation risk (±se) at different periods of night in different seasons.

Different lower case letters indicate significant difference at 0.05 level.

Table 5 The catch proportion of rodent in 2017, at Alxa Zuo Qi, Inner Mongolia, China.

The catch proportion of rodent (±se). Different lower case letters indicate significant difference among seasons at 0.05 level.

Rodent species	Catch rate / %	
	Spring	Summer	Autumn	
Dipus sagitta	1.523 ±0.553A	1.860 ±0.877A	0.595 ±0.298A	
Phodopus roborovskii	1.652 ±0.869	—	—	
Meriones meridianus	3.194 ±1.576A	2.009 ±0.902A	1.936 ±0.597A	
∑	4.846 ±0.583A	2.679 ±0.515B	2.531 ±0.299B	
Allactaga sibirica	5.234 ±0.5773A	3.646 ±1.225AB	1.486 ±0.298B	

The relationship between environmental factors and activity pattern in different seasons

The relationship between environmental factors and activity pattern in spring

The RDA results are displayed by an ordination diagram in which the dependent variable variables are depicted by blue arrows and impact factor by red arrows. The RDA biplot can be interpreted as follows: each blue arrow representing an impact factor determines a direction or axis in the diagram; red arrow representing activity time and frequency determine directions in the diagram. The correlations between activity time and frequency and impact factors are displayed by the angles of blue and red arrows. Arrows pointing in almost the same direction indicated a highly positive correlation, arrows oriented at right angles indicate nearly zero correlation, and arrows pointing in opposite directions indicate a highly negative correlation (Li & Kendrick, 1995). Activity time and frequency and impact factors with the longest arrows are the most important in the analysis; the longer the arrows, the more confident one can be about the inferred correlation (Braak & Prentice, 1988). Analysis of the relationships between different factors and activity time showed that the cosine value of the line segment representing activity time and temperature, food resource was positive. The cosine value of the line segment representing activity time and predation risk was 0, so there was no correlation between activity time, and perceived predation risk. The cosine value of the line segment representing activity time, wind speed, relative humidity, and intraspecific competition were negative. Thus, there were negative correlations between activity time, wind speed, relative humidity, and intraspecific competition. The line segment representing temperature and relative humidity was longer, so temperature and relative humidity had a more significant impact on activity time.

The analysis of the relationships between environmental factors and activity frequency showed that the cosine value of the line segment representing active frequency and temperature and intraspecific competition was 0, so there was no correlation. A positive correlation was found between activity frequency and predation risk, food resources and interspecific competition, and a negative correlation between activity frequency and relative humidity and wind speed. Among these factors, relative humidity had more significant impact on activity frequency, and this factor explained a larger proportion of variation in activity frequency.

Relative humidity was found to significantly affect the activity of Siberian jerboa in spring (RH F = 12.2, P = 0.002) (Fig. 4).

Figure 4 Ordination diagram showing the results of RDA analysis of different factors and activity time and frequency in spring.

The cosine value between activity time, activity freq uency and different variables represents the correlation between them. A positive cosine value indicates a positive correlation, and a negative value represents a negative correlation. The length of the line segment represents the magnitude of the factor’s explanation of activity time and activity frequency. AT: activity time; AF: activity frequency; T: temperature; WS: wind speed; RH: Air relative humidity; PR: perceived predation risk; FR: food resource; InterC: the relative population number of coexisting species; IntraC: the relative population number of Siberian jerboa.

The relationship between environment factors and activity pattern in summer

There were negative correlations between activity time and relative humidity, temperature, wind speed, food resources, interspecific competition, intraspecific competition, and perceived predation risk. Among these factors, the lines representing relative humidity and temperature were longer, indicating that their influence on the activity time was relatively more important.

There were negative correlations between activity frequency and each factor. Among them, temperature and relative humidity had a more significant explanatory value, and had greater impacts on activity frequency.

Temperature and relative humidity had significant impacts on the activity of this rodent species in the summer (T F = 11.4, P = 0.002; RH F = 29.4, P = 0.002; InterC F = 4.4, P = 0.028) (Fig. 5).

Figure 5 Ordination diagram showing the results of RDA analysis of different factors and activity time and frequency in summer.

The cosine value between activity time, activity frequency and different variables represents the correlation between them. A positive cosine value indicates a positive correlation, and a negative value represents a negative correlation. The length of the line segment represents the magnitude of the factor’s explanation of activity time and activity frequency. AT: activity time; AF: activity frequency; T: temperature; WS: wind speed; RH: Air relative humidity; PR: Perceived predation risk; FR: Food resource; Note: AT: activity time; AF: activity frequency; T: temperature; WS: wind speed; RH: Air relative humidity; PR: perceived predation risk; FR: food resource; InterC: the relative population number of coexisting species; IntraC: the relative population number of Siberian jerboa.

The relationship between environment factors and activity pattern in autumn

There were positive correlations between activity time and temperature and predation risk, food resources, and intraspecific competition. There were negative correlations between activity time and wind speed, and interspecific competition. There was no correlation between activity time and relative humidity. Among these factors, temperature, relative humidity, wind speed, and perceived predation risk had more considerable explanatory value, and their impact on activity frequency was more significant.

There were positive correlations between activity frequency and temperature and perceived predation risk. There was a negative correlation between activity frequency and relative humidity and wind speed and food resource.

Relative humidity, and perceived predation risk had significant impacts on the activity of this rodent species in the autumn (RH F = 6.5, P = 0.010; PR F = 33.5, P = 0.002) (Fig. 6).

Figure 6 Ordination diagram showing the results of RDA analysis of different factors and activity time and frequency in autumn.

The cosine value between activity time, activity frequency and different variables represents the correlation between them. A positive cosine value indicates a positive correlation, and a negative value represents a negative correlation. The length of the line segment represents the magnitude of the factor’s explanation of activity time and activity frequency. AT: activity time; AF: activity frequency; T: temperature; WS: wind speed; RH: Air relative humidity; PR: perceived predation risk; FR: food resource; Note: AT: activity time; AF: activity frequency; T: temperature; WS: wind speed; RH: Air relative humidity; PR: perceived predation risk; FR: food resource; InterC: the relative population number of coexisting species; IntraC: the relative population number of Siberian jerboa.

Discussion

There are costs and benefits associated with environmental and biotic demands, such that individuals usually tune their activities to the most favourable period in a day (Refinetti, 2008), and through this way individuals try to maximize their fitness when determining the proper time for basic survival and breeding activities. The Siberian jerboa had two peak periods in spring and summer, which were 21:00 -00: 00 and 02:00 to 04:00. The activity time and frequency in autumn were very low, and there were three activity peaks. Our results are similar to those of a previous study of the ecological habits of Siberian jerboa (Liang & Xiao, 1982). However, their research showed that the Siberian jerboa had a high intensity period of activity in September. This difference may be due to different conditions between the experimental sites (Dong, Hou & Yang, 2006), suggesting that the species is adaptable to different environments. The optimal response of an organism to change in its environment is to minimize the cost to it through some kind of adaptive response (Wootton, 1984). Activity peaks of Siberian jerboa were bimodal during the spring and summer, and trimodal during the autumn. Studies have shown that rodents change the number of peaks of activity depending on the temperature of the seasons (Erkert & Kappeler, 2004; Levy, Dayan & Kronfeld-Schor, 2007). The temperature of the region was lower in autumn than in summer, and reducing activities in the cooler autumn may minimize exposure in cold environments to reduce energy consumption (Cotton & Parker, 2000). However, the temperature in autumn was similar to that in spring, or was even slightly higher than in spring. The difference in activity peaks may be attributed to the various factors that affected the activities in different seasons.

Our results showed that the factors affecting activities were different between seasons. In spring, relative humidity affected activities; in summer, temperature, relative humidity, and interspecific competition affected activities; in autumn, wind speed, relative humidity, and perceived predation risk affected activities. Although the relationship between temperature and activity time was negative in summer, the fundamental mechanism was the same as in the other two seasons and each species had its optimal temperature range (Rezende et al., 2003). Temperatures were higher in summer than in spring and autumn, exceeding the jerboa’s temperature range. The relative humidity of different seasons had a negative impact on jerboa activities, which indicated that relative humidity was an important factor affecting the activity of this species. Other studies have shown that relative humidity promotes rodent activity in arid and semi-arid areas (Brodie, Formanowicz & Brodie, 1991), which was contrary to our findings. Studies have shown that the effects of humidity on animals vary, which may be due to the different ecological habits of each species (Zhang et al., 2006). The Siberian jerboa is a hibernating species (Zhou et al., 1992) and the activity of hibernating species is negatively related to humidity (Zhang et al., 2006). This difference in different seasons indicates that this species’ activity in different seasons was not affected by a single factor, rather it was affected by a combination of multiple factors and the variations of the elements in different seasons. These results support our hypothesis. Therefore, seasonal differences in the number of activity peaks can be attributed to the activity being affected by various factors throughout different seasons. Some studies have suggested that the number of peak periods is affected by temperature (Kei & Motokazu, 2017), but this is inconsistent with our results. We considered the influence of multiple factors while previous studies only considered the impact of single element (Ricardo, António & Pedro, 2011). There are many findings related to factors that influence animal activity patterns. Most studies of activity patterns have tended to focus on only one aspect and have failed to consider the relative importance of other factors (Shuai et al., 2014; Kei & Motokazu, 2017). Ecological and behavioural relationships between small mammals, especially rodents, are well-documented. It is reasonable to suspect that these factors may significantly affect the overall utilization of the local environment (Delany, 1972). Changes in animal activity patterns are adaptations to the general atmosphere. Therefore, it is reasonable to believe that changes in activity patterns may be influenced by multiple factors, not just one (Rant, 1978). Numerous factors must be taken into considered when analysing variation or change in animal activity patterns.

Research shows that temperature and relative humidity in different seasons play essential roles in influencing the activities of this rodent. In addition to factors that work together in different seasons under the influence of external abiotic factors, this rodent also responds to changes in biological factors. The risk of predation influences both activity pattern and habitat use (Werner, 1991; Berger & Gotthard, 2008). Food resources, perceived predation risk, and competition (interspecific and intraspecific competition) influenced activity levels in different seasons. The optimal foraging theory predicts that risk-taking decisions vary in response to perceived levels of threat (Emlen, 1996; Kelly, Cypher & Germano, 2020) and this was reflected in our results. Previous studies of other nocturnal mammals have shown that these animals reduce the risk of predation by restricting foraging activities or the duration of periodic activities (Gilbert & Boutin, 1991; Mastureh et al., 2017; Mansureh & Morteza, 2018).Siberian jerboa came out of hibernation in spring after a long hibernation period and required food to replenish their energy. The search for food resources is an important factor for triggering activities during this period. The increased need for food resources led to an increase in the time and frequency of activities to achieve the maximum use of food resources. The jerboa ignored the effect of perceived predation risk and intraspecific competition when seeking food resources and became risk-prone. Their foraging strategy during this season involved antipredator mechanisms and risk-proneness. In summer, when predation risk increases, the activity time and frequency of the jerboa decreased. The Siberian jerboa chose to avoid predation risk and competition. In summer, the adopted foraging strategy of the Siberian jerboa was risk-aversion and predator avoidance mechanisms, and reducing activity in the micro-habitat with high feeding pressure increased survival value (Clarks, 1983). In autumn, the jerboa prepared to enter hibernation, and needed to store energy for the hibernation period. At this time, food resources guided the activities of this species. The decrease in food resource led to a reduction of activity time and frequency. The need for food made them ignore predation risks and interspecific competition. The campaign foraging strategy of the season was the same as in spring, driven by the demand for food resources. The selection of food resources by the Siberian jerboa was diverse depending on the season, which indicates that it used the micro-habitat differently and selected habitat as an antipredator strategy. The demand for food in autumn encouraged the rodent to expand the species of plants that were selectively eaten, thereby allowing it to obtain more food resources at a level where the overall vegetation biomass of the habitat was not high. There was no significant difference in the amount of food resources between spring and summer but the species was found to take greater risks in autumn when the food resources were abundant. Food-deficient conditions generally cause animals to alter their behaviour from risk-aversion to risk-proneness, leading scholars to propose the risk-sensitive foraging theory (McNamara & Houston, 1990). Challenges in obtaining food resources lead Ochotona curzoniae to increase ground activity time to make full use of and protect their food resources (Zhang et al., 2005). Rodents reduced their exposure time and increased their activity frequency to reduce predation risk, which was one of the main countermeasures for adapting to high-risk environments (Yang et al., 2007). Animal have been shown to take risks to acquire resources when it is necessary for survival, even if the predation risk level increases (Barnard & Hurst, 1987; Helfman, 1984). This behaviour may explain why spring and autumn seasons promoted different behavioural strategies than summer; spring and autumn required greater demands for food resource acquisition. The adoption of this foraging strategy is driven by the demand for food resources, not by the amount of food resources. Thus, the need for food was an important influencing factor of overall activity during different seasons and animals possess the ability to integrate disparate sources of information about danger to optimize energy gain (Chelsea et al., 2019).

Conclusions

We determined that factors affecting activities were different among various seasons. In spring, relative humidity mainly affected activities. In summer, temperature, relative humidity, and interspecific competition mainly affected activities. In autumn, relative humidity and perceived predation risk mainly affected activities. The activity pattern of the Siberian jerboa was altered in different seasons. The Siberian jerboa had two similar peak periods in spring and summer, and there were three activity peaks in autumn with lower activity time and frequency. Various factors are known to affect animal activity at different levels. Abiotic factors (temperature, relative humidity and wind speed) acted on the daily activity level and affected the number of peak periods of activity in different seasons. The demand for food resources affected the level of activity throughout the seasons. The jerboa adapted different responses to predation risks and competition in different seasons according to the amount of food resources available.

Supplemental Information

Supplemental Information 1 Data analyzed

Each worksheet represents one type of data.

Click here for additional data file.

We would like to thank LN Ye, YN Li, QW Guo, DS Wen, YL Jin, and LL Li for their assistance in data collection.

Additional Information and Declarations

Competing Interests

Author Contributions

Data Availability

The authors declare there are no competing interests.

Yu Ji performed the experiments, analyzed the data, prepared figures and/or tables, authored or reviewed drafts of the paper, and approved the final draft.

Shuai Yuan, Heping Fu and Xiaodong Wu conceived and designed the experiments, authored or reviewed drafts of the paper, and approved the final draft.

Suwen Yang, Fan Bu and Xin Li performed the experiments, prepared figures and/or tables, and approved the final draft.

The following information was supplied regarding data availability:

The raw measurements are available as a Supplemental File.

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
