# Peer review of "Activity strategy and pattern of the Siberian jerboa (Orientallactaga sibirica) in the Alxa desert region, China"

_PeerJ, doi:10.7717/peerj.10996_

## Round 0.1 · original submission · Major Revisions

Hello, thank you for your submission.

It has been reviewed by two experts in the field and both felt the paper needed some additional work. Please address the reviewers' comments to the best of your ability and re-submit.

Reviewer 1 ·

Basic reporting

The manuscript should be largely improved in terms of expression and scientific writing. Some expressions are ambiguous or even misleading, and the writing sometimes lacks preciseness. I suggest the authors seek for help from a native English speaker or a language service company. Introduction on background is poorly organized and seems to be unfocused.

Experimental design

This paper describes an interesting seasonal pattern of activity pattern in a mainly nocturnal rodent. The authors then seek to associate these changes with several environmental factors using RDA. This research should fit within the scope of PeerJ and may deserve publication, but not in its current form. First, the Introduction is not well organized and it is not clear why the authors choose this topic or why they decide to use this system. It is thus difficult to evaluate whether this research helps to fill some important knowledge gaps. Some important details of methods are not clearly described. Please see my specific comments for more information.

Validity of the findings

The validity of the findings needs to be strengthened, as I have some methodological concerns. Some procedures need to be validated and clarified (e.g. using vigilance behavior as a surrogate for predation risk, using lures to attract jerboas, individual identification, and performing ANOVAs without a check for the suitability of data).

Additional comments

Specific Comments

Line 2: The term “strategy” is unclear. What type of strategy do you refer to? Foraging strategy? Habitat selection? Life history strategy? Use an appropriate word throughout the manuscript.
Line 13: Seems to be an invalid affiliation.
Line 25: The term “activity time” is ambiguous. Do you mean the duration of active period or the time point when the rodent is active?
Line 48: Interspecific competition also plays a role in shaping activity patterns.
Lines 100-102: The logic here is problematic. I agree that this species can be viewed as a habitat generalist or a wide-ranged species as you mentioned, but could it represent other generalists in terms of activity patterns? Generalists can have various activity patterns (diurnal, nocturnal or crespucular, et al.).
Line 110: Energy supply is always needed as long as the animal is alive.
Line 117: Rewrite this part as “may encourage jerboas to withstand higher predation risk”.
Lines 135-137: What about the abundance and activity patterns of these sympatric rodents? Would interspecific competition also affect activity pattern of jerboas?
Line 138: What about the activity patterns of these predators? Are they active in the same period as jerboas? This is important for your hypothesis associated with predation risk.
Line 144: What does the “grid” mean? A site? A grid of cameras? A trapping grid? How many cameras are placed in each grid?
Lines 147-148: You need to add some support (e.g. citations) for this idea. In my experience, five-toed jerboas may occasionally be active during daytime, and some mainly nocturnal rodents may become more diurnal as the weather gets colder. In this sense, it would thus be somewhat risky to solely focus your investigation on nighttime. It does not mean that your analyses are invalid, but you do need to consider this potential limitation, at least in Discussion.
Line 151: When did you place the lure? How many peanuts are used for a lure? Would the lure significantly modify the behavior of jerboas and thus affect your results?
Line 162: Did you adopt some procedures to mark individuals? How could you tell whether it is the same animal or not?
Line 165: “Sentinel behavior” generally occurs in cooperative species. It means that some individuals act as sentinels while other group mates keep on foraging, as can be seen in meerkats. For jerboas, “vigilance behavior” may be a more appropriate word.
Lines 165-167: Please clarify how to define this behavior. What types of physical action response?
Lines 170-172: Where are the climatic data collected? Could they reflect the local environmental conditions in your study sites?
Lines 177-179: It is somewhat problematic to directly use vigilance behavior as an indicator of predation risk. Actually, vigilance reflects a trade-off among many aspects, such as predation risk, interference competition, individual energetic state, and so on. For instance, suppose you detect a jerboa to be less vigilant during the early night, it does not necessarily mean that the actual predation risk is low. Maybe it is simply because that at this stage the jerboa is rather hungry, so energetic intake rather than antipredation is more important for him.
Line 185: How do you estimate the biomass of the plants? Did you cut them off and measure their dry weight?
Lines 210-211: You need to test first whether your data meets the requirements (e.g. normality and homogeneity of variance) for ANOVAs. If not, data transformation, GLM, non-parametric tests or Monte Carlo methods should be used instead.
Line 227: Specify the two degrees of freedom for each F statistic, and hereafter.
Line 232: Where is the data supporting this point?
Line 312: Actually, animals often have to be active during less favorable periods, due to competition, predation risk or other environmental stressors.

Fig. 2-3: What do the bars represent? S. E. or S. D.?

Reviewer 2 ·

Basic reporting

English should be improved. Some parts of the text are nor clear enough. Some terms are formally correct, but usually not used in professional literature.
Literature references are not sufficient.
The structure of the article is formally correct.
Theoretical background of the tested hypotheses is week. Results are interesting only as the natural history description of the species.

Experimental design

This is original research which is out of aims and scope of the journal.
Research questions are defined based on week theoretical background.
Data collection was performed to a high technical & ethical standard.
Methods described without sufficient details.

Validity of the findings

Presented results are interesting as the natural history description of the species.
Underlying data are provided; they are robust, statistically sound, & controlled.
Conclusions are well stated but not linked with original research question

Additional comments

The manuscript contains new data on seasonal patterns of daily activity of Siberian jerboa in Alashan desert of Inner Mongolia obtained with camera trapping technique. These data are interesting as the natural history description of the species. However, theoretical background of the paper is week in several aspects listed below:
1. Authors used a bait (peanut) and thus, they measured mainly foraging activity.
2. Authors does not describe in details what they considered as the “sentinel” behavior. It is necessary to provide description of the poses or the sequences of behavioral bouts that they attributed as alertness.
3. Authors estimated abundance of foraging plant species based on the set of cafeteria experiments, but they did not record used parts of plants. It is well-known that Siberian jerboa feed on green parts of plants, underground parts of plants, seeds and insects in approximately equal proportion (Shenbrot et al., 2008 - Jerboas. Mammals of Russia and adjacent regions. Science Publishers Inc, , Enfield, USA). Thus, their estimations of food abundance are not complete.
4. Authors did not provide the data on abundance of predators in the study area. In the absence of such data, alertness can be explained also by searching the partners for mating and locating flying insects for foraging.
5. Considering the time allocation for different types of activities, authors missed breeding activity, which is important at least in spring and beginning of summer.
6. Finally, it is not correct to consider the time allocation for different types of activities out of the frames of the theory of optimal foraging.
Introduction is written in a chaotic style. Most references to studies of mammalian activity patterns are not related to rodents in desert environments, whereas rodent activity in deserts has very specific patterns.
In discussion, there are no references to publications on activity patterns of other species of jerboas (summarized in Shenbrot et al., 2008).
Incorrect English and Latin names of the studied species. Authors used the name “five-toed jerboa” and “Allactaga sibirica”. Genus Allactaga s.lato was found to be polyphyletic and because of this was divided into three separate genera; the species authors studied belong to the genus Orientallactaga (Lebedev et al., 2013-- Zoologica Scripta 42: 231-249; Michaux & Shenbrot 2017 - Family Dipodidae (Jerboas). In: D.E. Wilson, T.E. Lacher, Jr. and R.A. Mittermeier (eds). Handbook of the Mammals of the World. Vol 7. Rodents II. Lynx Editions, Barcelona. Pp. 62-100). “Five-toed jerboa” is the name used for all species of the subfamily Allactaginae; correct English name for Orientallactaga sibirica is Siberian jerboa.

---

## Round 0.2 · Minor Revisions

Thank you for taking the time to make the requested revisions. They have been looked over by two expert reviewers. While both felt that you addressed the initial concerns, there are still some minor points that need clarifying.

Please address the minor points raised by reviewers and resubmit your manuscript.

Reviewer 1 ·

Basic reporting

In this revised manuscript, the authors have dealt with my opinions accordingly in most cases, which greatly improved the paper quality. However, in terms of language and grammar, this paper is still poorly written (I have listed some errors in specific comments). Moreover, many important data was absent, or not correctly stated. I strongly recommend the authors have the manuscript thoroughly checked and edited by a native English speaker, or (even better) a language editing company, to make this paper more readable.

Experimental design

This paper fits within Aims and Scope of PeerJ. The research question is interesting and relevant, and the methodology is appropriate in most cases.

Validity of the findings

Some important data is missing, or not correctly stated. Conclusions seem to be correct but some need to be identified as speculation.

Additional comments

General Comments

In this revised manuscript, the authors have dealt with my opinions accordingly in most cases, which greatly improved the paper quality. However, in terms of language and grammar, this paper is still poorly written (I have listed some errors below). Moreover, many important data was absent, or not correctly stated. I strongly recommend the authors have the manuscript thoroughly checked and edited by a native English speaker, or (even better) a language editing company, to make this paper more readable.

Specific Comments:

Line 16: It seems that a space was missing between “in” and “activity”.
Line 62: Add “main” before “factor”.
Line 73: Delete “been”.
Line 106: Add the Latin name after “jerboa”.
Line 174: Again, how could you tell whether they are different individuals of the same species, as you didn’t adopt any procedures to mark jerboas?
Lines 179-180: Grammatical errors.
Line 203: Add “the” before “three plots”. Check your manuscript carefully before resubmission, since such grammatical errors appear frequently throughout the paper.
Line 219: Write it as 95.80 ± 17.74 (Mean ± SD).
Line 292: The sentence “Considering…” is not a complete sentence.
Line 295: The sentence “the electronic chips…” should appear at Line 299. What type of chips did you use? PIT tags? The information on manufacturer and model is also needed. What about the weight of the chips?
Line 305: Delete “whether the leg rings are lost”.
Line 309: Use O. sibirica here and hereafter.
Line 337: Use “among” to replace “between”.
Lines 385-425: In this section, many important data is missing. For example, where is the data supporting the “negative correlations between activity time and relative humidity, temperature, wind speed…”? Lines 387-398 are also needed to be rephrased in a more concise way.
Line 438: Use “between” rather than “in”.
Line 503: Use “encourage” rather than “enable”.
Lines 507-508: Rephrase this sentence as “Why does it still choose to take risk in autumn, when food resources are abundant?”
Line 517: This sentence needs to be rephrased.

·

Basic reporting

Dear Editor in Chief
This paper is talking about the activity pattern of the Siberian Jerboa. The overall structure of the paper is suitable but it needs some minor revisions. Meanwhile there is some excellent related papers on Jerboas activity patterns in desert areas from Iran, which I proposed to authors and I wish they add them to the citations and use them in the discussion.
Anyway, my detailed comments are embedded in the manuscript word file.
Journal of Wildlife and Biodiversity 1(1): 33-36 (2017) by Arak University, Iran (http://www.wildlife-biodiversity.com/)
DOI: 10.22120/jwb.2017.27208
Nocturnal activity and habitat selection of Hotson Jerboa, Allactaga hotsoni Thomas, 1920 (Rodentia: Dipodidae)
I strongly suggest to use and cite these papers:

Journal of Wildlife and Biodiversity 1(1): 33-36 (2017) by Arak University, Iran (http://www.wildlife-biodiversity.com/)
DOI: 10.22120/jwb.2017.27208
Nocturnal activity and habitat selection of Hotson Jerboa, Allactaga hotsoni Thomas, 1920 (Rodentia: Dipodidae)
Mastureh Darabi, Zohreh Zeini, Abdolreza Karami, Ali Kaveh

1. Notes on habitat affinities of the Hotson's Jerboa Allactaga hotsoni Thomas, 1920 (Rodentia: Dipodidae) from Isfahan province, Iran
Mansureh Khalatbari; Morteza Naderi
Volume 2, Issue 1, Winter 2018, Pages 41-43
10.22120/jwb.2018.30219
http://www.wildlife-biodiversity.com/article_30219.html

Nocturnal activity of Iranian jerboa, Allactaga firouzi (Mammalia: Rodentia: Dipodidae)
Mahmoud-Reza Hemami 1 , Gholamreza Naderi 2 , Mahmoud Karami 3 , and Saeed Mohammadi 4

Experimental design

The experimental design is based on camera trapping methods which seems suitable but the authors can explain how the similar projects used this manner as well.

Validity of the findings

In my idea, Jerboas are very interesting animals in desert areas which can be use an ideal model for investigating on small mammals activity patterns.

Additional comments

I suggested two related papers which Authors can use in the discussion part.

---

## Round 0.3 · accepted · Accept

Thank you for making all of the revisions requested by the reviewers and addressing their points. Based on your revision, I think your paper can be accepted.